# The mediation effect of attitude on the association between knowledge and practices toward air pollution among commercial drivers and traders in South-Western Ghana: A cross-sectional study

Enoch Akyeampong[1,2☯*], Abdulzeid Yen Anafo[3☯], Jesus Kofi Asante[1‡], Isaac Kwabla Agbenyezi[1‡], Richard Amfo-Otu[4☯], Benson Owusu[5‡], Michael Affordofe[6‡], John Kofi Nyante[7‡], Maxwell Seyram Sunu[7‡]

1 Department of Occupational Health and Safety, Accra School of Hygiene, Korle Bu, Ghana, 2 Department of Environmental and Safety Engineering, Univerity of Mines and Technology, Tarkwa, Ghana, 3 Department of Mathematics and Statistics, University of Mines and Technology, Tarkwa, Ghana, 4 Department of Environment and Public Health, University of Environment and Sustainable Development, Somanya, Ghana, 5 School of Nursing and Midwifery, Central University, Miotso, Ghana, 6 Department of Environmental Health and Sanitation, Accra School of Hygiene, Korle Bu, Ghana, 7 Environmental Quality Unit, Environmental Protection Authority, Accra Central, Ghana

☯ These authors contributed equally to this work
‡ These authors also contributed equally to this work
* enochakyeampong@gmail.com

## Abstract

There are extant studies on the knowledge, attitudes and practices of air pollution in sub-Saharan Africa and Ghana, however, studies among commercial drivers and traders in transport stations in Ghana are sparse. This study examined the mediation effect of attitude on the association between knowledge and practice toward air pollution among two high-risk workers in the Greater Accra Metropolitan Area, Ghana. A population-based cross-sectional study was conducted among commercial drivers and traders between November 2023 and June 2024. A consecutive sampling technique was used to select 1011 participants (drivers [n = 619, 61.2%] and traders [n = 392, 38.8%]). A structured questionnaire was used, including questions on knowledge (9 items), attitude (7 items) and practices (5 items) toward air pollution. Statistical analysis (descriptive, hierarchical linear regression and partial least squares structural equation modelling) was performed using Stata 17/MP. The primary sources of air pollution information for drivers and traders were social media and radio. The PLS-SEM revealed a significant direct effect of knowledge for both drivers/traders (β = 0.23/0.26, p < 0.001). Moreover, attitude partially mediated the relationship between knowledge and practices, with a direct significant effect observed for drivers/traders (β = 0.10/0.02, p < 0.001). HLR further confirmed that knowledge strongly predicted protective practices for both drivers (β = 0.23, p < 0.001) and traders (β = 0.26, p < .001. Higher educational level consistently predicted better knowledge and

**Data availability statement:** All data files are available from the Harvardverse database (https://doi.org/10.7910/DVN/QCBTA9).

**Funding:** The author(s) received no specific funding for this work.

**Competing interests:** The authors have declared that no competing interest exist.

protective practices but old age had an inverse relationship with protective behaviour toward air pollution. Although a cross-sectional design precludes causality, attitude partially mediated the association between knowledge and practices, but the effect was more substantial among drivers than traders. The findings have implications for social and traditional media education and attitudinal change campaigns to effectively reduce air pollution exposure risk among these high-risk occupational groups.

## Introduction

As Ghana undergoes rapid urbanization, like many other African nations, it has become critical to influence policy that guides businesses to adopt pragmatic and comprehensive steps to address the causes of air pollution which are connected to their occupations [1–3]. Air pollution poses a significant public health challenge in Ghana, with 28,000 Ghanaians dying prematurely each year due to its effects [4,5]. While there are extant studies on the knowledge, attitudes and practices (KAP) of air pollution in high income countries(HICs) [6–12], only a few are known in low- and middle-income countries (LMICs), particularly in sub-Saharan Africa (SSA) [13–19] and Ghana [5,20]. These studies consistently reveal disparities in knowledge and preventive behaviors, with populations in HICs often exhibiting a firmer grasp of causes and health effects of air pollution. In contrast, attitudes toward air pollution tend to vary less predictably by income level, suggesting that cultural, social, and occupational contexts may influence how individuals translate knowledge into action. However, similar studies among commercial drivers and traders in transport stations in Ghana are notably not known.

The exponential growth in the number of motor vehicles due to the growing population in urban areas leads to a further increase in ambient air pollution. The situation is aggravated in Ghana, where the reliance on second-hand vehicles for transportation business is on the rise in urban areas. Of the estimated 3.2 million cars registered in Ghana, the Greater Accra Metropolitan Area (GAMA) has the highest (1.2 million) [21,22]. Most of these vehicles, which operate in transportation stations, are old and have poor maintenance histories, emitting dangerous gases and heat that harm commuters. The influx of commuters to these transportation stations creates business opportunities for traders. Commercial drivers and traders operating in transportation stations are routinely exposed to vehicular emissions, dust, and other airborne pollutants, making them particularly susceptible to the adverse health effects of ambient air pollution. The increase in cardiorespiratory infections in Ghana may be due to this interaction. Understanding the KAP of these occupational groups is essential for designing effective, context-specific public health interventions.

Although national strategies such as air quality monitoring and public awareness campaigns led by the Environmental Protection Authority (EPA) and Ministry of Health of Ghana have been introduced to address the burden of air pollution, evidence on how these policies influence risk perceptions or protective behaviors among at-risk occupational groups [23,24] is limited. Moreover, whether attitude

mediate the relationship between environmental knowledge and the adoption of protective practices remains unclear. Therefore, this study, aimed to assess the levels of KAP concerning air pollution among commercial drivers and traders in selected transportation stations in South-Western Ghana. Specifically, it examined whether attitude mediates the relationship between knowledge and practice, and explored how respondents perceived sources of air pollution relate to sources of information. This study sought to fill critical evidence gaps and support the development of targeted, evidence-based strategies to promote safer occupational environments and reduce air pollution-related health risks in Ghana's urban transport sector.

## Literature review and hypothesis development

**Knowledge and attitude toward air pollution.** Knowledge about air pollution is essential for developing attitudes and shaping behavior. Extant studies with contradictory results highlight the importance of knowledge in influencing public perception of air quality. For example, public awareness and support for air pollution mitigation measures in Malaysia have been examined [25]. KAP was assessed using a cross-sectional design, and the findings revealed that individuals with higher knowledge levels exhibited more positive attitudes toward adopting effective air pollution mitigation strategies. Also, Wang et al. [10], in a cross-sectional study, examined the relationship between knowledge and attitude about air pollution and respiratory health among parents in Shanghai, China. The results showed that knowledge about the health effects of air pollution is positively associated with supportive attitudes toward protective measures. Similarly, in the United States, Mirabelli et al. [26] used a cross-sectional design to assess the impact of the air quality index on public attitudes. It was revealed that knowledge significantly shaped proactive attitudes toward protective measures, accentuating the importance of awareness messages. In addition, Qin et al. [8] used a cross-sectional study to assess the knowledge and perception of air pollution in Ningbo, China and found high attitudes. Nevertheless, the knowledge was low among the elderly, less educated and some occupations. However, a survey done in Dhaka, Bangladesh, to investigate the KAP regarding particulate matter pollution found that while higher levels of knowledge are associated with improved attitudes, differences in knowledge due to socio-economic difficulties hamper significant attitudinal shifts [13]. Furthermore, Shupler et al. [3] global estimation study of $PM_{2.5}$ exposure from household air pollution, emphasized the critical need for targeted interventions to address knowledge gaps. A mixed-methods study on industrial air pollution in rural Kenya by Omanga et al. [15] found that partial awareness significantly weakens attitudes toward environmental risks, stressing the importance of knowledge in shaping attitudes. Amponsah et al. [1] studied Ghana's transition from reactive to proactive environmental measures and showed the systemic challenges in fostering proactive attitudes due to knowledge gaps. Similarly, Gadzekpo et al. [20] used a cross-sectional study to assess KAP on climate change mitigation and practices in Ghana. They found high attitudes but low knowledge in mediating climate change messages. These studies imply that good levels of knowledge are crucial for fostering favorable attitudes toward air pollution. Despite the disparities in the findings, it is hypothesized that;

H1: Attitude has a direct and positive effect on knowledge of commercial drivers and traders about ambient air pollution in commercial drivers and traders at GAMA

**Practices and attitude toward air pollution.** The impact of practices on attitudes is multi-layered, often shaped by experiential learning. The interaction of these variables mostly generates conflicting outcomes anchored on geographical and socio-economic differences. For example, a descriptive cross-sectional study among medical students in Colombia was conducted by Rendon-Marin et al. [6] to assess their KAP toward air pollution. The results indicated that hands-on practices, such as participating in air quality monitoring, positively and significantly influenced attitudes toward environmental health. In Malaysia, a mixed methods approach was used to examine the association between KAP and haze pollution [11]. The results showed that proactive measures, such as the use of face masks, nurtured favorable

attitudes, supporting the idea of a reciprocal relationship between practices and attitudes. In Accra, Ghana, Odonkor and Mahami [5] used a critical survey design to understand the public perception of air pollution. The results from the study revealed that community-driven practices, including participation in clean-up campaigns, significantly enhanced attitudes toward environmental stewardship. Similarly, West et al. [16] used a citizen science approach in Nairobi, Kenya, to demonstrate that community engagement in visualizing PM pollution could enhance knowledge and attitudinal change. However, systemic challenges, such as resource limitation, often erode the sustainability of these practices. Kemba et al. [14] used a cross-sectional design to assess the KAP among tertiary education students in Windhoek, Namibia. The results showed that practical engagement in pollution reduction activities was limited due to institutional and systemic challenges despite moderate levels of knowledge and positive attitudes. The conflicting results from the reviewed studies underscore the interplay between experiential practices and the development of sustainable attitudinal change toward air pollution. Thus, we hypothesize that

> H2: Practices have a direct and positive effect on attitude of commercial drivers and traders toward ambient air pollution at GAMA

**Knowledge and practices.** Studies have shown that attitudes significantly influence the knowledge-practice association. Nevertheless, the findings of these studies could be more consistent. This mediation effect of attitude on the relationship between knowledge and practice was studied using the Knowledge-Attitude-Behaviour model in Taiwan [9]. The study, which involved 1000 participants, applied structural equation modeling and found that attitude significantly mediated the relationship between knowledge and practice. Conversely, Marin et al. [12] cross-sectional study among 800 Colombian high school students showed partial mediation effects. The study showed that knowledge positively influenced attitudes, but the translation of attitudes into practices was mostly disrupted by external factors such as limited access to clean technologies. Furthermore, a comparative study on the impact of environmental education on attitudes and behaviors among Hungarian high school and university students found that knowledge effectively translated into practices through positive attitudes [27]. Flanagan et al. [17] used a longitudinal design to assess the effects of an indoor air pollution awareness campaign. The campaigns improved attitudes, but cultural and economic factors hampered their impact on practices. Similarly, a cross-sectional study which examined traffic police KAP in Nairobi, Kenya, found that weak attitudes, even among knowledgeable people, constrained the adoption of effective practices against vehicle emissions [19].

The reviewed studies and conclusions infer that while attitudes are significant mediators in the relationship between knowledge and practice, their effectiveness is hampered by economic and cultural factors and geographical differences. We, therefore, hypothesize that

> H3: Practices have a direct and positive effect on knowledge and attitude among commercial drivers and traders toward ambient air pollution at GAMA

## Materials and methods

### Study setting

This study was conducted in the urban areas of GAMA (Fig 1) of Ghana's Greater Accra Region (GAR). GAR is the smallest region in terms of land area (3245 km²) but has the highest population density and is the most populous. GAMA is the economic, administrative and industrial hub of GAR and is home to 4.3 million people. The region has several informal transportation stations, many of which are regulated by the Ghana Private Road Transport Union (GPRTU) with elected executives. Two main lorry stations (Kaneshie and Kwame Nkrumah Circle) were chosen as the study sites. These sites

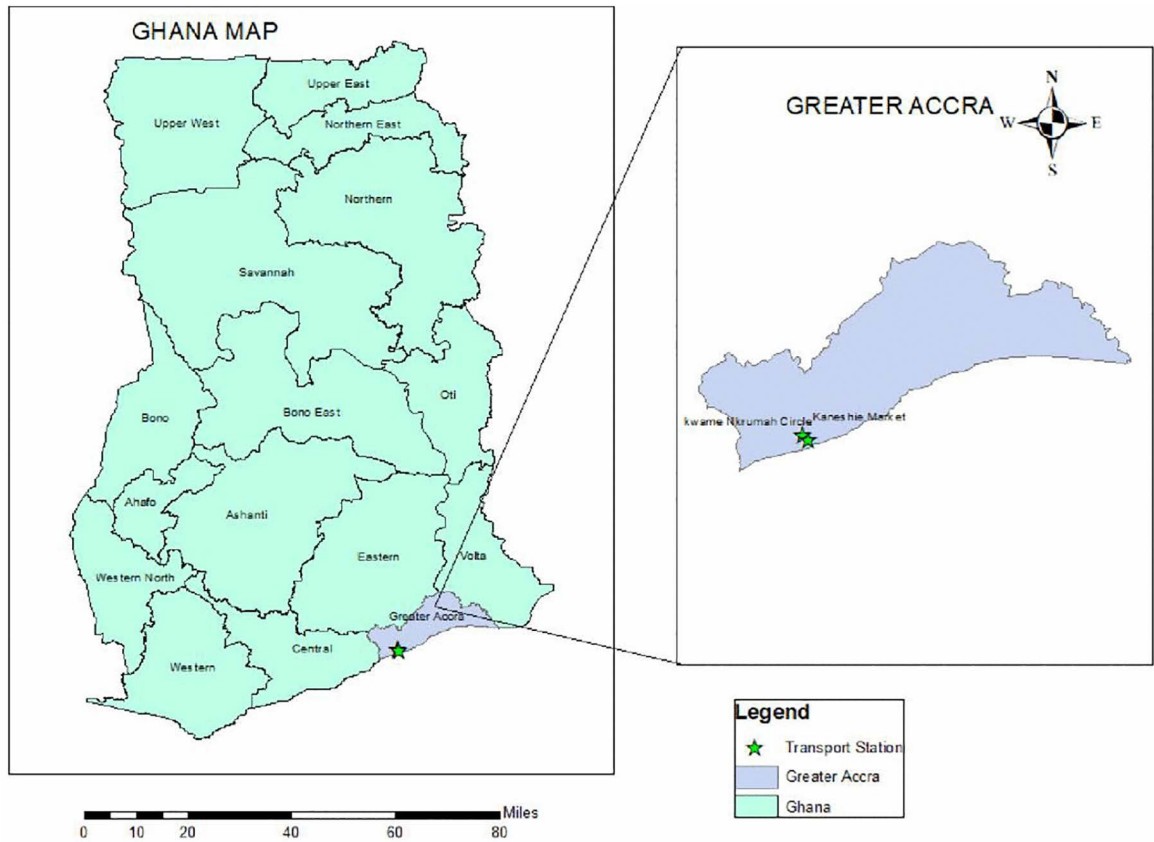

**Fig 1. Map showing data collection points in GAMA.** *Left: national map with Greater Accra highlighted (gray). Right inset: district map of Greater Accra with sampled transport stations shown in green stars. (Basemap: OpenStreetMap).*

serve as transport hubs as tro-tro (mini-buses) and motorcycle and taxi stations for inter- and intra-city travel. The outer lanes along the main roads also serve as loading and off-loading sites for mini-buses, taxis and motorcycles. Furthermore, these transport stations serve as trading hotspots. Most traders spend most of the day trading because human congestion provides fertile grounds for sales. Most vehicles registered in the study sites are second-hand and emit exhaust fumes. Nonetheless, the economic benefits of their operations expose them to high volumes of PM pollution, which has negative health implications for traders and commercial drivers.

## Study design and participants

This population-based cross-sectional study was conducted in GAMA from 21st November 2023–22nd June 2024. The study points included Ghana's rainy (March – October) and dry (November – February) seasons. This study was conducted among traders and commercial drivers with irregular work timelines. We used a finite population formula based on an estimated population of 7000 and 6000 at Kwame Nkrumah Circle and Kaneshie respectively [28] to arrive at a sample size of 1011 traders and commercial drivers ([Z-score for 95% confidence interval of 1.96, margin of error = 0.05, expected prevalence of health outcome = 50% and 15% non-response rate). A description of the participant selection procedure is presented in Fig 2. Consequently, a consecutive sampling technique was used. The study protocol was approved by the Korle Bu Teaching Hospital Institutional Review Board (KBTH-IRB/000295/2023), and all participants signed written informed consent before participation in the study.

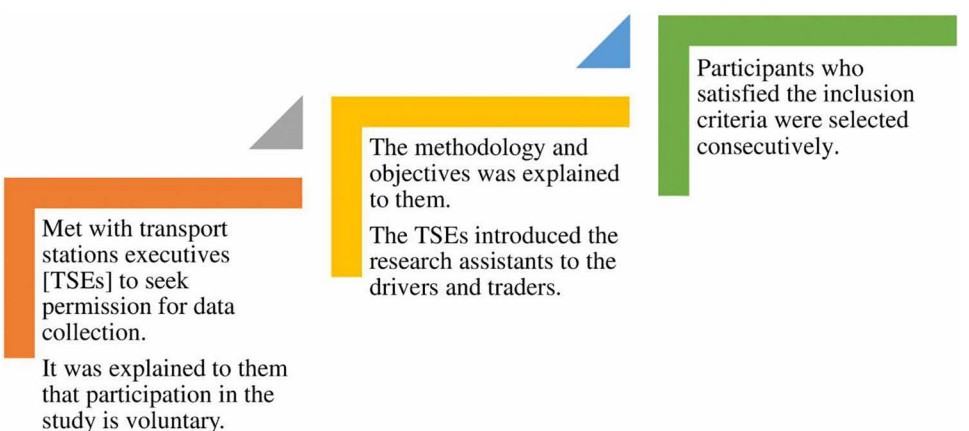

**Fig 2. Sampling procedure. A schematic diagram representing how permission was sought and participants selected for questionnaire administration.**

### Inclusion and exclusion criteria

The inclusion criteria for participation in the study were being an adult (≥18 years old for both drivers and traders), having worked for at least four years in the transport stations, spend 5 hours or more in the transport station per day, possess a valid driver's license and drives either a mini-bus (tro-tro), long haul bus, motorcycle or taxi (drivers only) and operates a shop or stall (traders only). Participants who were very busy and declined consent to participate were excluded.

### Questionnaire

Trained research assistants conducted face-to-face interviews using standard questionnaires. The questionnaire included questions related to personal information, including sex (male or female), age (18±35, 36±45, 46±56, and ≥ 60 years), education, length of activity per day, duration of work and comorbidity. The questions about education were categorized as no formal education, primary, JSS/JHS/Form 4, SSS/SHS/Vocational/technical, tertiary education (degree, diploma, certificate) and graduate degree (master's degree, PhD). Furthermore, there were questions related to occupation factors, which included information on the occupation type (driver [e.g., taxi, mini bus, long haul bus, motorcycle driver [okada, pragyie]), traders, and length of work activity per day (4-5h vs ≥ 6h). Questions related to medical history of non-communicable diseases were included. The questionnaire also captured knowledge, attitudes and practices toward air pollution. The items in the questionnaire were adapted from multiple sources: knowledge [13], attitude and practices [8]. Knowledge of air pollution was measured using nine items, including sources of air pollution, health effects, and protection practices. The questionnaire on attitude toward air pollution had seven items, including air quality in the workplace, improving air quality in the workplace and access to information about air quality. Five items related to practices of protection against air pollution were measured, including measures taken against air pollution, protection measures taken and reducing the length of working hours. Each participant spent about 25 minutes completing the questionnaire.

### Statistical analysis

Data were entered into Microsoft Excel and imported to Stata 17/MP for analysis. Analyses were conducted at three levels. First, background data were analysed using frequencies and percentages stratified by drivers and traders. Pairwise comparisons and simple regression models of the dependent variable (practice), mediator variable (attitude), and independent variable (knowledge) were conducted, and p-values were provided. Cronbach's alpha (α ≥ 0.7) was used to test the internal consistency, and item-total correlation was used to discard the weak items (r < 0.3). Low-reliability scales were

refined by removing or revising inconsistent items. The reliability coefficients were knowledge (α = 0.79, n = 7); attitude (α = 0.70, n = 4) and practices (α = 0.78, n = 5). Linear regression explored the relationship between drivers and traders and KAP. Next, we performed mediation analyses using the partial least squares structural equation modeling (PLS-SEM) to estimate mediation effects with 5,000 bootstrapping samples based on recommendations (RMSEA<0.08, GFI, >0.9, CFI>0.09 and TLI>) provided by Awang et al [29, 30]. The causal steps proposed by Byaes were used to test for mediation. Lastly, we modeled the association between KAP using hierarchical linear regression (HLR) while controlling for age, educational level, and sex. Estimates are presented as coefficients and confidence intervals. The significance level was $p < 0.05$.

## Results

### Background characteristics

This study involved 1,011 participants consisting of 619 (61.2%) drivers and 392 (38.8%) traders (Table 1). About 34.6% of the respondents were aged between 46 and 55 years, and more than half (56.5%) were males. Close to 41% of the respondents attained senior high-level education at the time of the study, and most of the drivers (64.8%) and traders (52.1%) have been working for 4–5 years. Approximately 55% of the respondents spend 7–9 hours working each day.

### Sources of air pollution information

Fig 3 shows the results of the sources of air pollution information among drivers and traders. Mass media, radio and television, (49.2%) was the primary source of information about air pollution, highlighting its importance in public health campaigns. Nevertheless, significant disparities in information sources was observed between both occupational groups. While drivers demonstrated significantly higher reliance on mass media (63.2%) and social media and internet use (40.9%), traders reported lower user reliance, suggesting that their differing media consumption patterns could lead to gaps in access to air quality information.

### Correlation analysis of KAP toward air pollution

The correlation between KAP was examined for drivers and traders in Kaneshie and Kwame Nkrumah Circle (Figs 4–6). Knowledge is a stronger driver for good practices for both drivers (β = 0.23, p < 0.001) and traders (β = 0.22, p < 0.001) (Fig 4), suggesting that educational campaigns should be used to drive public health interventions because increased knowledge fostered a more favorable attitude toward air pollution for both drivers/traders (β = 0.20/0.22, p < 0.001) (Fig 5). However, the weaker association between attitude and practices, especially among drivers (β = 0.22, p < 0.001) (Fig 6) show a significant attitude – behavior gap toward air pollution. This suggests that the leadership must make desired practices easier to adopt by integrating them into the transport stations infrastructure. Although traders showed a higher knowledge- attitude relationship and drivers had a higher attitude, the interaction between these constructs was statistically not significant.

### Mediation analysis of the relationship between knowledge, attitude and practice

The structural path analysis (Fig 7) confirms that knowledge has significant total effect on practice for both drivers (β = 0.26, p < 0.001) and traders (β = 0.24, p < 0.001). Attitude functions as a statistically significant positive mediator in this relationship for both groups, however, the mediation strengthvaries significantly. Attitude is a significant mediator for drivers (direct effect: β = 0.20, p < 0.001), explaining the influence of a large portion of knowledge on drivers attitude. The mediation for traders, while significant, is minimal and indirect (β = 0.02, p < 0.001). This suggests that the impact of knowledge on practice is almost entirely direct for traders, where the mediating role of attitude is negligible, whereas for drivers, the impact is partly fostered through the cultivation of positive attitude.

**Table 1. Background characteristics of study participants.**

| Characteristics | Total N = [1,011] n (%)[c] | [a]Driver [n=619] 61.2% | [b]Trader [n=392] 38.8% |
|---|---|---|---|
| **Sex** | | | |
| Male | 571 (56.5) | 339 (54.8) | 323 (59.2) |
| Female | 440 (43.5) | 160 (40.8) | 280 (45.2) |
| **Age group (Years)** | | | |
| 18–25 | 80 (7.9) | 32 (5.2) | 48 (12.2) |
| 26–35 | 130 (12.9) | 88 (14.2) | 42 (10.7) |
| 36–45 | 313 (30.9) | 205 (33.1) | 108 (27.6) |
| 46–55 | 350 (34.6) | 212 (34.2) | 138 (35.2) |
| 56–65 | 117 (11.6) | 68 (11.0) | 49 (12.5) |
| 66–75 | 21 (2.1) | 14 (2.3) | 7 (1.8) |
| **Educational level** | | | |
| No formal education | 120 (11.9) | 58 (9.4) | 62 (15.8) |
| Primary | 67 (6.6) | 43 (6.9) | 21 (6.1) |
| JHS[d] | 303 (30.0) | 172 (27.8) | 131 (33.4) |
| SHS[e] | 414 (41.0) | 286 (46.2) | 128 (32.7) |
| Tertiary | 107 (10.6) | 60 (9.7) | 47 (12.0) |
| **Duration of work (in years)** | | | |
| 4 - 5 | 577 (57.1) | 322 (52.1) | 254 (64.8) |
| ≥6 | 434 (42.9) | 296 (47.9) | 138 (35.2) |
| **Length of activity per day (in hours)** | | | |
| 5–6 | 183 (18.1) | 122 (19.7) | 61 (15.5) |
| 7–9 | 556 (55.0) | 341 (55.1) | 215 (54.9) |
| ≥10 | 272 (26.9) | 156 (25.2) | 116 (29.6) |
| **Comorbidity** | | | |
| Yes | 730 (72.2) | 433 (69.9) | 297 (75.8) |
| No | 281 (27.8) | 186 (30.1) | 95 (24.2) |

a: driver is someone who drives taxi, trotro, long haul buses or rides okada (motorcycle)

b: trader is someone who either sells merchandize or retail or wholesales non-perishable items

c: freqency(percentage)

d: Junior High School

e: Senior High School

### Hypothesis testing using hierarchical linear regression

Three HLR examined predictors of knowledge, attitude and practices towards air pollution among drivers and traders while controlling for age, sex and education (Table 2). In Model 1, which examined knowledge scores, the control variables of age, education, and sex accounted for 6% of the variance ($R^2 = 0.06$) and occupation (p=0.016) and sex (p<0.001) were associated with increased attitude whereas increasing age showed a negative association (p=0.016). After adding knowledge in model 1[δδ], the explanatory power increased to 7.0% ($\Delta R^2 = 0.07$), with attitude emerging as a strong positive predicator of knowledge (p<0.001), supporting the hypothesis that attitude directly influences knowledge toward ambient air pollution. Furthermore, the knowledge score (Model 2), and the control variables alone explained 21.0%% of the variance ($R^2 = 0.21$) with education (p<0.001), sex (p<0.001), and occupation (p<0.001) being significant predictors. The addition of attitude score in Model 2[δδ], significantly improved the model fit ($\Delta R^2 = 0.22$), indicating that

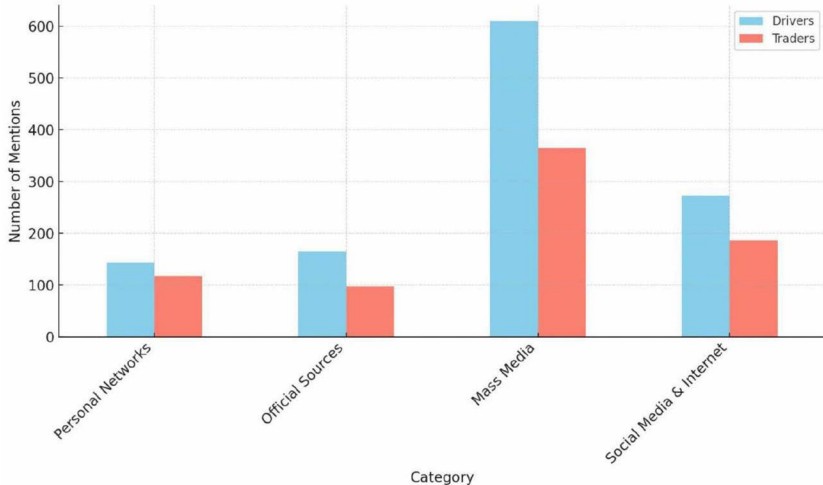

**Fig 3. Sources of air pollution information among drivers and traders. (NOTE: Multiple responses, n = 1931)**.

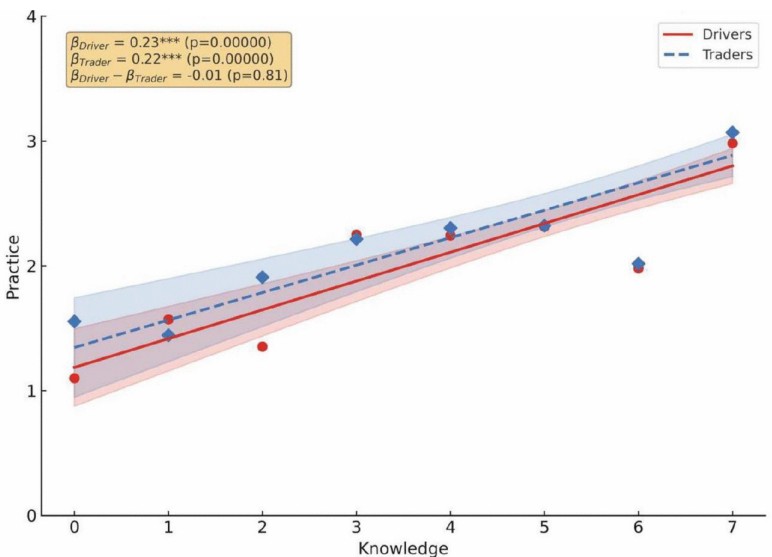

**Fig 4. Pearson correlation analysis between drivers and traders' knowledge and practice toward air pollution.**

knowledge is a strong predictor of attitude (p < 0.001). This shows that knowledge mediates the effect of education on attitude, whereas the control variables alone showed limited influence on attitude toward ambient air pollution. The practice score (Model 3), showed that control variables explained 8.0% of the variance in protective behavior ($R^2 = 0.08$). Sex (p < 0.001) predicted positive protective behaviors but increasing age was associated with poorer protective behaviours (p < 0.001). The variance improved slightly ($\Delta R^2 = 0.09$) after adding knowledge score (model 3[δδ]) which is a significant predictor of protective behaviors (p = 0.002). When we added the attitude score (model 3[δδδ]), we found an increase in the variance ($\Delta R^2 = 0.10$), with the attitude score showing a positive significant effect (p < 0.001) in improving protective behaviors between drivers and traders. This is consistent with the hypothesis that practices have a direct and positive effect on drivers and trader's knowledge and attitude toward ambient air pollution at GAMA.

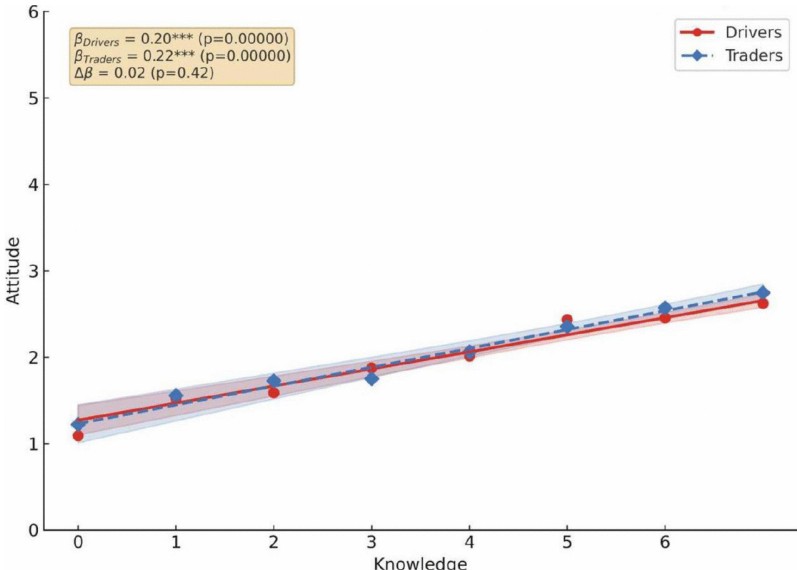

**Fig 5. Pearson correlation analysis between drivers and traders' knowledge and attitude toward air pollution.**

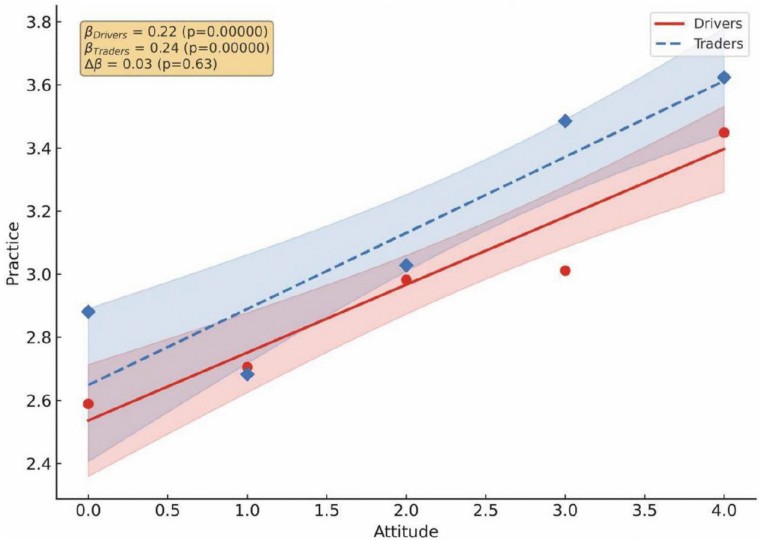

**Fig 6. Pearson correlation analysis between drivers and traders' practice and attitude toward air pollution.**

## Discussion

### Main findings

We examined the mediating effect of attitude in the relationship between knowledge and practice toward air pollution among commercial drivers and traders with three hypotheses, all of which have been accepted in our results. Our study provides evidence that a higher level of knowledge significantly predicts better protective practices toward air pollution. Attitude exerted a partial mediation effect on this relationship. Television/radio and social media were the primary sources

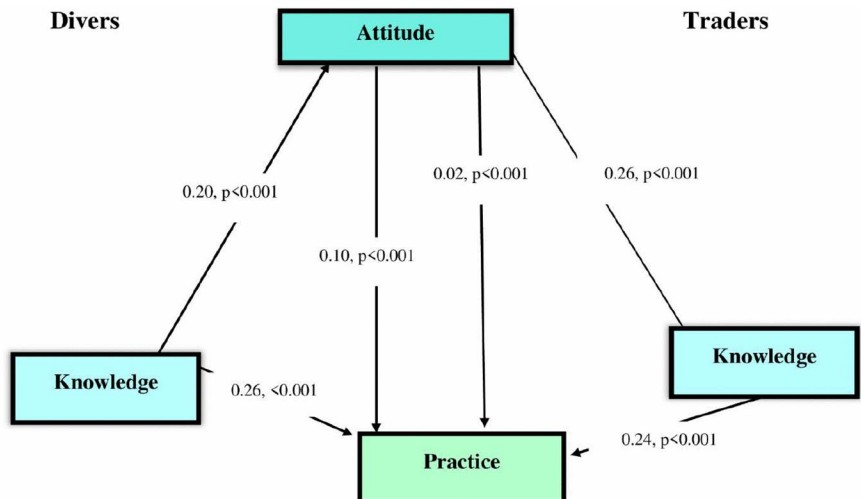

**Fig 7. SEM standardized paths coefficient with attitude as the mediator in the relationship between knowledge and practices toward air pollution.**

of air pollution information, and a higher educational level consistently predicted better knowledge and protective practices, but old age had an inverse relationship with protective behavior. Although statistically significant, drivers showed superior KAP toward air pollution than traders.

## Comparison with previous studies

The results of our study highlight the significant association between knowledge and practices toward air pollution. Specifically, a strong positive association was observed between knowledge and practice for both drivers/traders ($\beta = 0.23/0.22$, $p < 0001$), translating into improved protective practices toward air pollution among both groups of workers, thus, the hypothesized positive association between practices and knowledge appears to be in place. Notably, these workers are aware of the dangers air pollution poses to their wellbeing. This accentuates the importance of continual and heightened education and awareness about the importance of adopting and using protective equipment to mitigate the harmful effects of ambient air pollution, consistent with findings from Malaysia [25], Shanghai, China [8,10] and United States [26]. Other studies have reported contradictory findings. For example, in their study on KAP regarding PM pollution in Dhaka, Majumder et al. [13] identified that differences in socioeconomic status weaken the knowledge – attitude relationship. Similarly, Gadzekpo et al. [20] reported high attitudes but low knowledge in mediating climate change messages in their study on KAP on climate change mitigation and practices in Ghana. Strengthening educational campaigns is essential to mitigating the harmful effects of air pollution, however, bridging the gap between awareness and action for workers in high risk occupations requires combining such interventions with affordable protective equipment and proactive workplace policies.

Traders showed a moderately positive association with protective practices compared with drivers. This implies that in shaping attitudes among drivers and traders, practical and hands-on engagements are important, undoubtedly due to their direct exposure to air pollution from exhaust fumes, smoke from burning and dust. This approach has proven to be true in studies conducted in Ghana and Columbia. The study by Rendon-Marin [6] study among medical students in Columbia proved that practical engagement in air quality monitoring improved the attitudes of medical science students, the study by Odonkor [5] study showed that communal engagement in clean-up exercises in Accra stimulated stronger pro-environmental attitudes. This reinforces the belief that active participation enhances attitudinal change. Nevertheless, Kemba et al. [14] found that institutional constraints slowed students active participation in air pollution monitoring

**Table 2. Hierarchical linear regression analysis on KAP toward air pollution in GAMA.**

| | Attitude | | Knowledge | | Practice | | |
| --- | --- | --- | --- | --- | --- | --- | --- |
| | Model 1 | Model 1[�] | Model 2 | Model 2[ﬂ] | Model 3 | Model 3[ﬀ] | Model 3[ﬃ] |
| Variables | β (95%CI), p-value | β (95%CI), p-value | β (95%CI), p-value | β (95%CI), p-value | β (95%CI), p-value | β (95%CI), p-value | β (95%CI), p-value |
| Control Variable | | | | | | | |
| Age | −0.03(−0.05, −0.01), 0.016* | −0.02(−0.05, −0.01), 0.035* | −0.04(−0.06, −0.01), 0.003* | −0.03(−0.06, −0.01), 0.008* | −0.07(−0.09, 0.04), 0.001** | −0.06(−0.09, −0.04), 0.001** | −0.06(−0.09, −0.03), 0.001** |
| Education | 0.07(0.05, 0.09), 0.001** | 0.05(0.03, 0.08), 0.001** | 0.18(0.15, 0.20), 0.001** | 0.17(0.15,0.20), 0.001** | 0.08(0.06,0.11), 0.001** | 0.07(0.04,0.09), 0.001** | 0.06(0.03,0.09) 0.001** |
| Sex | 0.05(−0.02, 0.12), 0.177 | 0.03(−0.04,0.10), 0.373 | 0.17(0.09,0.25), 0.001** | 0.16(0.09,0.24), 0.001** | 0.07(−0.02,0.13), 0.113 | 0.05(−0.03,0.14), 0.242 | 0.05(−0.04,0.13), 0.282 |
| Occupation | 0.09(0.02, 0.16), 0.016* | 0.08(0.003,0.15), 0.040* | 0.13(0.05,0.22), 0.001** | 0.12(0.04,0.20), 0.003* | 0.04(−0.05,0.13), 0.382 | 0.02(−0.06,0.11), 0.577 | 0.01(−0.07,0.10), 0.746 |
| | | | | | | | |
| Predictors | | | | | | | |
| Knowledge | – | 0.10(0.04,0.15), 0.001** | – | – | – | 0.11(0.04,0.17), 0.002* | 0.09(0.03,0.16), 0.001** |
| Attitude | – | – | – | 0.12(0.05,0.19), 0.001** | – | – | 0.14(0.06,0.21), 0.001** |
| | | | | | | | |
| Model Summary | | | | | | | |
| Constant | −0.13(−0.35, 0.09) | −0.09(−0.31, 0.13) | −0.38(−0.62, −0.13) | −0.36(−0.61, −0.11) | 0.34(0.08,0.61) | 0.38(0.12,0.65) | 0.39(0.13,0.66) |
| F-statistic | 15.3(<0.001) | 14.8(<0.00) | 65.9(<0.00) | 55.7(<0.000) | 20.79(<0.000) | 18.7(<0.000) | 18.0(<0.000) |
| R-square | 0.06 | 0.07 | 0.21 | 0.22 | 0.08 | 0.09 | 0.10 |
| Adjusted R² | 0.05 | 0.06 | 0.20 | 0.21 | 0.07 | 0.08 | 0.09 |

β: Coefficient

CI: Confidence interval

p-value notation: $p < 0.05$*; $p < 0.001$**

Model 1 (Attitude score): includes only control variables (age, educational level, sex, and occupation)

Model 1[ﬂ]: include all control variables and knowledge score to predict attitude score

Model 2 (Knowledge score): all control variables (age, educational level, sex, and occupation) were included

Model 2[ﬂ]: include all control variables and attitude score to predict knowledge score

Model 3 (Practice score): all control variables (age, educational level, sex, and occupation) were included

Model 3[ﬂ]: predicts practice score using the control variables and the knowledge scores

Model 3[ﬃ]: predicts practice score using the control variables and the knowledge and attitude scores

irrespective of their moderate knowledge and positive attitude, which resonates with the weaker association among drivers. The success of drivers rest on daily sales; this coupled with structural and limited resources limitations in transport stations, compresses the association between practice and attitude. Furthermore, West et al. [16] emphasized that citizen science and community engagement approaches can improve attitudinal change toward air pollution, yet systemic

challenges, such as resource limitation, often erode the sustainability of these practices. This agrees with our findings which shows that drivers stronger association confirms that continuous environmental engagement effectively sustains practices unlike the periodic practices among traders.

Furthermore, facemasks were preferred among traders (62.7%), and drivers preferred wearing hats (43.4%) as protective measures against air pollution (S1 Table). While these practices demonstrate consciousness and attitude of protection against the harmful effects of air pollution, loose-fitting masks (surgical masks) and improvised devices such as brimmed hats, do not provide adequate protection as the N95 respirators do [31,32]. The cost disparities between the surgical mask and the N95 respirators, despite the enforcement laxity in the use of nose mask post COVID-19, make the surgical mask the preferred choice. However, the surgical mask fails to provide adequate levels of protection 99% of the time [33–35]. This supports the notion that surgical masks are not designed to provide an airtight seal of protection. Therefore, air particulate particles can bypass the seal and enter the respiratory tract during normal breathing, causing potential harm to users [31]. This highlights the need to disseminate the right information to workers in high-risk occupations (such as drivers and traders) on the need not to re-use their surgical mask, importantly, the N95 respirators must be used among these high-risk occupations for maximum protection against inhaling harmful ambient air pollutants.

The mediating effect of attitude in the relationship between knowledge and practice is a key highlight of our result. As hypothesized, of the two occupations studied, the attitude toward air pollution, which partially moderates the association between knowledge and practice, was higher among drivers than among traders (indirect effect: 0.10 vs 0.02, $p < 0.001$). This indicates that the impact of drivers' knowledge on their protective behaviors was partly through attitude. The partial mediation effect supports the knowledge-attitude-behavior model [9]. The influence of external factors found in Colombia [12], such as restricted access to protective services, may be reflected in the weaker mediation among traders. As successfully shown in air quality monitoring programs, our findings highlight the necessity of interventions that actively seek to change attitudes through community participation and experiential learning in addition to improving knowledge [12]. However, our findings highlight the need for attitude-focused strategies to overcome contextual barriers that impede behavior change, especially for vulnerable populations such as traders who ply their trade in high-risk areas like transport stations.

Additionally, in our study, there were significant differences in the KAP results by demographic characteristics such as occupation, age, sex, and education. As demonstrated in an earlier Accra study, the stronger mediation effect among drivers than among traders most likely reflects variations in risk perception influenced by direct exposure to PM emissions from vehicles [5]. The differences in education and gender are consistent with the findings of other Ghanaian studies [5,20], indicating that environmental health awareness remains uneven. Given the inconsistent findings from Nairobi, the negative correlation between age and preventive practices suggests generational disparities in environmental participation which requires further research [16]. These differences highlight the significance of tailored solutions that consider age-specific communication strategies, educational attainment, and occupational risk.

Consistent with studies from Ghana [20] and Malaysia [25], radio, television, and social media are prominent as primary information sources in this study, highlighting their potential as instruments for efficient environmental health communication. However, concerns about possible misinformation [3] suggest that these campaigns or awareness creation efforts should be designed in collaboration with regulators and health authorities, such as the EPA, the Ministry of Communication, Digitalization and Innovation, and Ghana Health Services. Despite high awareness levels, a gap still exists between knowledge and practice, which echoes Amponsah [1] criticism of reactive policy-making and reflects larger structural issues in Ghana's environmental governance. This emphasizes the critical need for intersectoral interventions that integrate education with regulatory actions, such as enforcement of emissions standards, regular maintenance and adopting cleaner alternative like electric vehicles, particularly for informal workers (including commercial drivers and traders in transport stations) who experience comparable neglect throughout SSA [18].

## Methodological considerations

Our study has several advantages. Although there are studies on KAP in Ghana, to the best of our knowledge, this is the first study to use mediation analysis to examine KAP towards air pollution among two high-risk occupational groups in transport stations. Control variables in the HLR were determined based on their statistical significance ($p < 0.05$) with the change in attitude/behaviour toward air pollution [8,30–32]. Apart from the high participation rates which reduced selection bias, the two locations where the survey was conducted were homogenous across several demographic variables and revealed clearer and immediate differences between the two groups. The use of PLS-SEM and HLR models mitigated any variances in the results across the demographics. Some limitations must be considered in interpreting the findings of our study. The cross-sectional design restricts causal interpretation because it does not allow for the establishment of temporal relationships among knowledge, attitude, and protective practices. Reliance on self-reported data introduces risks of recall bias and social desirability bias, which may cause participants to overstate their knowledge or compliance with protective measures. The generalizability of the findings is also limited because the research was conducted at only two transportation stations in South-Western Ghana, excluding other vulnerable groups such as children, pregnant women, and informal workers outside the transport sector. While occupational homogeneity of the population reduced variability, it limited the exploration of broader socio-economic or institutional influences. Although the KAP instrument was adapted and contextually validated, it lacked thorough psychometric evaluation, which could affect the interpretation of attitudinal measures.

## Conclusion

This study highlights the relationship between knowledge, attitudes, and protective behaviors concerning air pollution within two high-risk jobs commercial drivers and traders, at transport stations in South-Western Ghana. The results indicate that attitudes partially mediate the association between knowledge and protective behavior, with a stronger effect observed in drivers than in traders. This supports the relevance of the KAP model in occupational health environments. Importantly, while higher education and mass media facilitate knowledge and promote positive actions, challenges such as job demands, lack of access to protective gear, and socioeconomic barriers hinder the complete application of knowledge into practice.

Future studies should utilize longitudinal and intervention-based designs to determine causal pathways and assess the efficacy of behavior-change strategies. Additionally, future studies can use AI to predict the KAP toward air pollution. Broadening the focus to include other vulnerable groups such as children, pregnant women, and market-based food vendors will enhance the representativeness of the findings and lead to more inclusive policy responses. Ghana can bolster its resilience against the increasing public health risk of urban air pollution by connecting knowledge to action through evidence-based interventions.

## Supporting information

**S1 Table. Protective measures used against air pollution among drivers and traders.**
(DOCX)

## Acknowledgments

We are grateful to the managers of the transport stations (Kaneshie and Kwame Nkrumah Circle) and the leaders of the traders for granting us permission and participating in the study.

## Author contributions

**Conceptualization:** Enoch Akyeampong.

**Data curation:** Enoch Akyeampong, Abdulzeid Yen Anafo, Jesus Kofi Asante, Isaac Kwabla Agbenyezi, Benson Owusu, Richard Amfo-Otu, Michael Affordofe, John Kofi Nyante, Maxwell Seyram Sunu.

**Formal analysis:** Abdulzeid Yen Anafo.

**Investigation:** Enoch Akyeampong.

**Methodology:** Enoch Akyeampong, Richard Amfo-Otu.

**Project administration:** Enoch Akyeampong.

**Resources:** Enoch Akyeampong, Jesus Kofi Asante, Isaac Kwabla Agbenyezi, Benson Owusu, Michael Affordofe, John Kofi Nyante, Maxwell Seyram Sunu.

**Supervision:** Enoch Akyeampong.

**Writing – original draft:** Enoch Akyeampong.

**Writing – review & editing:** Abdulzeid Yen Anafo, Isaac Kwabla Agbenyezi, Benson Owusu, Richard Amfo-Otu, Michael Affordofe, John Kofi Nyante, Maxwell Seyram Sunu.

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
