## [Decision Letter · Decision Letter 0]

8 Sep 2025

Thank you for submitting your manuscript to PLOS ONE. After careful consideration, we feel that it has merit but does not fully meet PLOS ONE’s publication criteria as it currently stands. Therefore, we invite you to submit a revised version of the manuscript that addresses the points raised during the review process.

We have received the reviewer feedback on our manuscript. The overall assessment is not so negative, but several revisions are required before the paper can be accepted. Below is a summary of the main points. Please check your login for any attached review reports.

Please submit your revised manuscript by Oct 23 2025 11:59PM.  If you will need more time than this to complete your revisions, please reply to this message or contact the journal office at plosone@plos.org . A rebuttal letter that responds to each point raised by the academic editor and reviewer(s). You should upload this letter as a separate file labeled 'Response to Reviewers'.A marked-up copy of your manuscript that highlights changes made to the original version. You should upload this as a separate file labeled 'Revised Manuscript with Track Changes'.An unmarked version of your revised paper without tracked changes. You should upload this as a separate file labeled 'Manuscript'.

We look forward to receiving your revised manuscript.

Kind regards,

Jigneshkumar Pramodbhai Desai

Guest Editor

PLOS ONE

Journal Requirements:

Additional Editor Comments (if provided):

- Improve the figure quality and work upon all suggestions of reviewers. 

Reviewer #1:

- Improve language and figure quality

- Compare your findings with existing standard reports as well as experimental data

- Inclusion-Exclusion criteria must be discussed in detail

- Include AI based predictions from the dataset.

Reviewer #2:

This paper investigates a significant but less researched area by assessing the knowledge, attitudes, and practices (KAP) of commercial drivers and traders regarding air pollution in Ghana. The research is well-justified with a large sample size and regression and structural equation models (SEM) for mediation impact assessment. The ethical clearance and access to data have been well-discussed and thereby enhance transparency.

Nonetheless, a few issues need to be addressed before acceptance of the manuscript. The hypotheses at times conceptually blur (attitude → knowledge, etc.), and would be improved by more explicit justification or rewriting in accordance with traditional KAP/KAB frameworks. The modified questionnaire lacks psychometric validation (Cronbach's alpha, etc.), and so generates doubts concerning reliability. Statistical reporting also has room for improvement: p-values should consistently be formatted, SEM fit indices reported in full, and the use of PLS-SEM justified.

Also, the analysis is repetitive of results instead of critically evaluating them, and would benefit from greater emphasis on policy implications, for instance, cheap protective gear or enforcement of emissions. The figures and tables similarly require more precise labels and presentation. Clearing these issues will enhance clarity, rigor, and impact. In general, I strongly suggest few revisions.

Reviewers' comments:

Reviewer's Responses to Questions

**Comments to the Author**

1. Is the manuscript technically sound, and do the data support the conclusions?

Reviewer #1: Partly

Reviewer #2: Partly

2. Has the statistical analysis been performed appropriately and rigorously?

Reviewer #1: Yes

Reviewer #2: Yes

3. Have the authors made all data underlying the findings in their manuscript fully available?

Reviewer #1: Yes

Reviewer #2: Yes

4. Is the manuscript presented in an intelligible fashion and written in standard English?

Reviewer #1: No

Reviewer #2: Yes

Reviewer #1: Improve language and figure quality

Compare your findings with existing standard reports as well as experimental data

Inclusion-Exclusion criteria must be discussed in detail

Include AI based predictions from the dataset.

Reviewer #2: This paper investigates a significant but less researched area by assessing the knowledge, attitudes, and practices (KAP) of commercial drivers and traders regarding air pollution in Ghana. The research is well-justified with a large sample size and regression and structural equation models (SEM) for mediation impact assessment. The ethical clearance and access to data have been well-discussed and thereby enhance transparency.

Nonetheless, a few issues need to be addressed before acceptance of the manuscript. The hypotheses at times conceptually blur (attitude → knowledge, etc.), and would be improved by more explicit justification or rewriting in accordance with traditional KAP/KAB frameworks. The modified questionnaire lacks psychometric validation (Cronbach's alpha, etc.), and so generates doubts concerning reliability. Statistical reporting also has room for improvement: p-values should consistently be formatted, SEM fit indices reported in full, and the use of PLS-SEM justified.

Also, the analysis is repetitive of results instead of critically evaluating them, and would benefit from greater emphasis on policy implications, for instance, cheap protective gear or enforcement of emissions. The figures and tables similarly require more precise labels and presentation. Clearing these issues will enhance clarity, rigor, and impact. In general, I strongly suggest few revisions.

**Do you want your identity to be public for this peer review?** For information about this choice, including consent withdrawal, please see our Privacy Policy

Reviewer #1: **Yes: ** Dr Juhi Saxena

Reviewer #2: No

---

## [Author Response · Author response to Decision Letter 1]

12 Dec 2025

Response to Editors Comments

Comment 1: Please ensure that your manuscript meets PLOS ONE's style requirements, including those for file naming

Response 1: Thank you for this observation. We have revised our manuscript to meet PLOS One’s style requirement. The changes can be found in the uploaded file name.

Comment 2: Your ethics statement should only appear in the Methods section of your manuscript. If your ethics statement is written in any section besides the Methods, please move it to the Methods section and delete it from any other section. Please ensure that your ethics statement is included in your manuscript, as the ethics statement entered into the online submission form will not be published alongside your manuscript.

Response 2: We have deleted the ethics statements from the declaration section of our manuscript and maintained same in the methods section only. Additionally, our ethics statement with corresponding approval credentials are included in our manuscript.

Comments 3: We note that Figure 1 in your submission contain [map/satellite] images which may be copyrighted. All PLOS content is published under the Creative Commons Attribution License (CC BY 4.0), which means that the manuscript, images, and Supporting Information files will be freely available online, and any third party is permitted to access, download, copy, distribute, and use these materials in any way, even commercially, with proper attribution. For these reasons, we cannot publish previously copyrighted maps or satellite images created using proprietary data, such as Google software (Google Maps, Street View, and Earth). For more information, see our copyright guidelines: http://journals.plos.org/plosone/s/licenses-and-copyright.

Response 3: We appreciate your observation and the options provided to ensure academic honesty. We were unable to obtain permission from the original copyright holder to publish the figure (map) under the CC BY 4.0 license, hence, we opted to supply a replacement figure that complies with the CC BY 4.0 license. For our replaced map, we have included source information and declared that it for illustrative purposes only

Comment 4: Thank you for providing this information. Could you please clarify what assets from CartoDB were used, if any? Are they free to access or were they purchased?

Response 4: We accessed GIS maps. We used a free trial access; however, we have decided against citing CartoDB and citing OpenStreetMap, which is free to use under an open license consistent with CC BY 4.0 license.

Comment 5: Improve the figure quality and work upon all suggestions of reviewers.

Response 5: the quality of the figures has been improved significantly and all suggestions from reviewers responded to in a systematic manner.

Comment 6: Please confirm that all assets sourced from CartoDB have now been removed from your submission.

Response 6: All maps and spatial figures in this manuscript were created with OpenStreetMap as the base map. No CartoDB sourced assets have been used in our submission.

Authors response to Reviewers Comments

Reviewer 1

Comment 1: Improve language and figure quality

Response 1: Thank you for taking time to read our manuscript and the observations made with regards to the quality of language. We have read the entire manuscript again and sought external review to ensure that the language quality is improved. Additionally, we have improved the quality of our figures for clarity.

Comment 2: Compare your findings with existing standard reports as well as experimental data

Response 2: We appreciate your observation and suggestion. We have improved the results presentation by not merely reporting the numbers but by evaluating them as have been reported elsewhere by other researchers (see pages 13 -14).

Comment 3: Inclusion-Exclusion criteria must be discussed in detail

Response 3: Although the inclusion and exclusion criteria are clearly defined under study design and participants, we have delineated the inclusion and exclusion criteria and discussed in detail to make it clear.

Comment 4: Include AI based predictions from the dataset.

Response 4: thank you for insightful suggestion to incorporate AI based predictions. This is an important and interesting idea for future research. However, our methodological scope is focused on establishing structural relationships and mediation pathways. Introducing AI based predictions could shift the focus of our study from explanation to prediction, an approach reserved for a subsequent study. We have added a sentence for future research to acknowledge this fascinating direction.

Reviewer 2

This paper investigates a significant but less researched area by assessing the knowledge, attitudes, and practices (KAP) of commercial drivers and traders regarding air pollution in Ghana. The research is well-justified with a large sample size and regression and structural equation models (SEM) for mediation impact assessment. The ethical clearance and access to data have been well-discussed and thereby enhance transparency.

Comment 1: The hypotheses at times conceptually blur (attitude → knowledge, etc.), and would be improved by more explicit justification or rewriting in accordance with traditional KAP/KAB frameworks.

Response 1: Thank you for observation and suggestion. We acknowledge the error of the presentation of our findings, hence, we have corrected the error by rewriting in accordance with traditional KAP frameworks. In figure 5, therefore, it now reads knowledge attitude.

Comment 2: The modified questionnaire lacks psychometric validation (Cronbach's alpha, etc.), and so generates doubts concerning reliability.

Response 2: We appreciate your suggestion about the psychometric validation. We have performed an initial Cronbach alpha on the KAP, which revealed poor internal consistency. Upon review, it was determined that the preliminary dataset contained data entry and coding errors. The psychometric was repeated with the validated dataset, which showed good psychometric properties, for example, knowledge (a=0.79), attitude (a=0.70) and practices (a=0.78). All reported results are based on the corrected dataset (see page 11)

Comment 3: Statistical reporting also has room for improvement: p-values should consistently be formatted.

Response 3: Thank you for your observation and suggestion. We noticed that Table 2 did not report the p-values. We have corrected this by adding the p-values to ensure consistent format in presentation.

Comment 4: The analysis is repetitive of results instead of critically evaluating them, and would benefit from greater emphasis on policy implications, for instance, cheap protective gear or enforcement of emissions.

Response 4: We take notice of this suggestion. We have improved the presentation of the results from repeating the findings to evaluating the findings with emphasis on policy implications.

Comment 5: The figures and tables similarly require more precise labels and presentation.

Response 5: We have improved the clarity of the figures and improved the labels of both the Tables (page 17) and Figures for clarity.

---

## [Decision Letter · Decision Letter 1]

29 Dec 2025

The mediation effect of attitude on the association between knowledge and practices toward air pollution among commercial drivers and traders in South-Western Ghana: a cross-sectional study

PONE-D-25-35486R1

Dear Dr. Enoch Akyeampong and all,

We’re pleased to inform you that your manuscript has been accepted for the publication. Please find attached comments if any that required while submit final manuscript. 

Kind regards,

Jigneshkumar Pramodbhai Desai

Guest Editor

PLOS One

Additional Editor Comments (optional):

Authors have addressed all the comments and improved the manuscript. 

Reviewers' comments:

Reviewer's Responses to Questions

**Comments to the Author**

Reviewer #1: All comments have been addressed

Reviewer #2: All comments have been addressed

---

## [Editor Report · Acceptance letter]

PONE-D-25-35486R1

PLOS One

Dear Dr. Akyeampong,

I'm pleased to inform you that your manuscript has been deemed suitable for publication in PLOS One. Congratulations! Your manuscript is now being handed over to our production team.

Kind regards,

on behalf of

Dr. Jigneshkumar Pramodbhai Desai

Guest Editor

PLOS One